# Relationship between the TGFBR1 Gene and Molar Incisor Hypomineralization

**DOI:** 10.3390/jpm13050777

**Published:** 2023-04-30

**Authors:** Laura Georgina-Pérez, David Ribas-Pérez, Alexandra Dehesa-Santos, Asunción Mendoza-Mendoza

**Affiliations:** 1Department of Stomatology, Universidad de Sevilla, 41080 Seville, Spain; 2Department of Clinical Dental Specialities, Universidad Complutense de Madrid, 28040 Madrid, Spain

**Keywords:** molar incisor hypomineralization, genetic relationship, TGFBR1 gene

## Abstract

Background: Molar Incisor Hypomineralization Syndrome (MIH) is a problem of increasing incidence that represents a new challenge in the dental treatment of many of the children we see in our dental offices. Understanding the etiology of this syndrome (still unknown) will help us to prevent the appearance of this process. Lately a certain genetic relationship has been suggested in the syndrome. The aim of the present study was to explore the relationship between activation of the TGFBR1 gene and the development of MIH, as recent studies suggest that there may be an association in this regard. Materials and Methods: The study sample consisted of 50 children between 6–17 years of age with MIH, each with at least one parent and a sibling with or without MIH, and a group control of 100 children without MIH. The condition of the permanent molars and incisors was evaluated and recorded based on the criteria of Mathu-Muju and Wright. Saliva samples were collected after washing and rinsing of the oral cavity. Genotyping was performed with the saliva samples for the selection of a target polymorphism of the studied gene (TGFBR1). Results: The mean age was 9.7 years (SD 2.36). Of the 50 children with MIH, 56% were boys and 44% girls. The degree of MIH was predominantly severe (58%), with moderate and mild involvement in 22% and 20% of the cases, respectively, according to the classification of Mathu-Muju. The allelic frequencies were seen to behave as expected. The logistic regression analysis aimed to relate each polymorphism to the presence or absence of the factors. These results were inconclusive, with no evidence suggesting an alteration of the TGFBR1 gene to be related to the appearance of MIH. Conclusions: Within the limitations posed by a study of these characteristics, it can be affirmed that no relationship has been found between the TGFBR1 gene and the appearance of molar incisor hypomineralization.

## 1. Introduction

In the 1970s, dentists of the Swedish National Health System reported an increase in the number of cases of idiopathic hypomineralization of the permanent first molars and incisors. They described a hitherto unusual phenomenon in which some children attended in the dental office of the Swedish Dental Public Health Service with very well demarcated enamel lesions, different from dental caries, and with an unknown etiology. In addition, these enamel defects were very difficult to treat and to clean properly due to a very marked sensitivity. Furthermore, the materials available in those years for obturations were limited to silver amalgam in the posterior sector, so that the final restoration of these teeth with hypomineralization resulted in what has come to be called atypical restoration with a very limited survival time due fundamentally to fractures in the restored teeth. [1,2]. Over 20 years went by until the clinical condition was finally referred to as molar incisor hypomineralization (MIH) [3]. Karin Weerhijm first introduced the term in the workshop of the European Academy of Pediatric Dentistry (EAPD) in the year 2003, defining a syndrome characterized by hypomineralization of systemic origin of one or more permanent first molars and of some permanent incisors.

The diagnostic criteria established by the EAPD currently do not differ much from those established in 2003, and since 2010 include at least one affected first permanent molar (with or without incisors affected), the existence of demarcated opacities in the enamel, the possibility of posteruptive breaks in the affected enamel, and high clinical sensitivity in the affected teeth. This makes treatment complex, and the EAPD also cites atypical restorations in the affected MIH teeth and sometimes early extractions of these teeth as diagnostic criteria [4].

In order to clarify the etiology of the disorder, and since MIH is a problem of the enamel, we would need to understand the different factors that can affect enamel formation (amelogenesis), and more specifically enamel formation of the permanent first molars and permanent incisors. The most plausible explanation is that MIH is caused by an interaction of processes that intervene in amelogenesis starting in the early stages of life, probably from birth to 4–6 months of age [5], and spanning the period of up to 3–4 years of age [6].

Consensus remains lacking despite the many studies on the etiology of MIH. Attempts have been made to relate the disorder to prenatal, perinatal, and postnatal causal factors, based on longitudinal surveys involving different numbers of children, and many retrospective studies have been published in this regard [7,8,9]. Epidemiologically, MIH is found all over the world, and although the current globalization phenomenon may exert an influence, there appear to be no geographical or sociocultural circumstances capable of establishing a clear correlation between habits or environmental factors and the appearance of the syndrome [10,11,12,13].

In view of the above, research continues in an attempt to clarify the etiology of MIH, and in this regard attention has recently focused on the possibility of genetic influences combined with epigenetic or environmental factors that could lead to enamel defects in susceptible individuals [14]. In this context, Jeremias et al., in 2013, reported the possibility that certain genes implicated in enamel formation could contribute to the appearance of MIH. Indeed, taking into account that enamel formation is under genetic control, it would be reasonable to assume that genetic variations could alter amelogenesis. In the same way that certain genetic disorders secondary to mutations are related to different types of amelogenesis imperfecta, such phenomena could also be extrapolated to MIH [12,15].

More recent studies have explored the relationship between genetic polymorphisms of certain genes (specifically the TGFBR1 gene) involved in host immune response and MIH [16]. The present study was thus carried out to explore the relationship between activation of the TGFBR1 gene and the development of MIH, as these more recent studies suggest that there may be an association in this regard.

## 2. Materials and Methods

### 2.1. Type of Study and Population

A population-based genetic association study was carried out in abidance with the consolidated standards of reporting trials of 2010. The study sample consisted of 50 children between 6–17 years of age with MIH, and who had at least one parent and a sibling with or without MIH, seen in the pediatric dentistry clinic of the Dental School of the University of Seville (Spain). A series of 100 children without MIH were included as a control group.

The sample size was calculated taking into account the parameters of the study population, statistical power, and the prevalence of the disorder in the population.

The statistical power (1 − β) to detect the genetic association with the sample analyzed was determined using the TDT Power Calculator software and its value was 80%. A correction value of α = 0.002 adjusted for multiple testing, applying the Bonferroni correction, and a disease prevalence of 12% in the population studied was established based on epidemiological data published in Spain in 2015 [17]. 

### 2.2. Inclusion and Exclusion Criteria

Inclusion criteria: Children between 6–17 years of age, with at least one molar presenting structural alterations due to MIH (mild, moderate, or severe), and with one parent and a sibling with or without MIH who were willing to collaborate in the study.

Exclusion criteria: Patients with any other type of enamel defect, congenital syndromes, systemic diseases, or who were wearing fixed multibrackets.

### 2.3. Clinical Examination and Questionnaire

Prior meetings were held to clarify diagnostic concepts of MIH, and an intra- and inter-observer Cohen’s kappa concordance index of over 0.80 was obtained (acceptable). The condition of the permanent molars and incisors was evaluated and recorded based on the criteria of Mathu-Muju and Wright [18] (Appendix A).

The entire indexed tooth was kept moist during the examination in order to discard opacification attributable to excessive drying, and the size of the lesion was taken into account. Likewise, a previously validated questionnaire exploring prenatal, perinatal, and postnatal aspects was delivered to the parents or caregivers of the diagnosed children.

The clinical examination was carried out in the pediatric dentistry clinic of the Dental School of the University of Seville (Spain), using the enamel lesions evaluation protocol. A flat no. 5 mirror and 11.5 probe recommended by the World Health Organization (WHO) were used for the exploration, carried out by two dentists calibrated in MIH under the criteria of the European Academy of Pediatric Dentistry (IAP) and based on the Mathu-Muju scale (mild, moderate, or severe). Informed consent was obtained from all the participants in the study, which was approved by the Ethics Committee of the University of Seville (Ref. no. 1052-n-19).

### 2.4. Sample Collection and Genetic Analysis

Saliva samples were collected after washing and rinsing of the oral cavity. A 5 mL sample was obtained from each patient and placed in a 10 mL Falcon tube stored on ice for transport to the Molecular Genetics Laboratory of the University of Seville.

With this material, the instructions given by the manufacturer for this genetic analysis were followed. The tube was first centrifuged at a minimum of 3000× *g* to ensure that the material was located at the bottom of the tube. TE was added to reach the required final concentration of 10 ng/µL. Samples were vortexed briefly and incubated at 50 °C for 20 min. After brief vortexing and centrifugation, the Gene Fragments sample was amplified if its size was ≤1 kb, using a high-fidelity polymerase, limiting the cycles to 12 or less to avoid sequence mutations due to amplification errors. For larger fragments no amplification was performed.

Genotyping was performed using SNP Browser 4.0 software (Applied Biosystems Foster City, CA, USA) for the selection of a target polymorphism of the studied gene (TGFBR1). Each single nucleotide polymorphism (SNP) was genotyped with real-time polymerase chain reaction (PCR) using the TaqMan^®^ 5’-exonuclease (Thermo Scientific, MA, USA) allelic discrimination test and 7500 real-time PCR system (Applied Biosystems, Foster City, CA, USA). The reactions were carried out in an end-volume of 20 μL containing 2 μL of DNA at a concentration of 10 ng/μL, 0.96 μL of each TaqMan test reagent, 9.61 μL of TaqMan Genotyping Master Mix, and 7.43 μL of water without DNA.

The final samples (called GBlocks) could be the homozygous controls for the two variants. GBlock1: Homozygous GG; GBlock2: Homozygous AA. From the results, a graph was obtained where the G variant corresponds to blue fluorescence samples and the A variant to green fluorescence samples. The intermediate fluorescence of Heterozygous AG was plotted in red. Also represented were samples without DNA (in gray) that were used as negative control, and samples in magenta that were uncertain samples, not recognized by any of the three groups, possibly because of the quality of the DNA.

### 2.5. Statistical Analysis

The family-based associations were analyzed as a global group (all families with MIH), which in turn was divided into subgroups according to the severity of MIH (mild, moderate, or severe).

The parent and offspring genotypes were compared to determine the transmitted alleles versus the non-transmitted alleles, using the transmission disequilibrium test (TDT) [19,20]. This test was implemented according to the PLINK v1.07 application [21]. The frequencies of rare alleles (minimum frequency of alleles), the presence of Mendelian errors and the Hardy–Weinberg equilibrium were also determined for this population using the Gene-Calc tool. Statistical significance was considered for *p* < 0.05.

Gene–gene interaction between the polymorphisms of the immune response genes and genes related to amelogenesis was assessed taking into account the previous results of the analysis of polymorphism of the genes related to enamel formation described by Busanelli in 2019 [16]. The gene–gene interaction analyses were made observing transmission of the marker alleles (one corresponding to immune response and the other to tooth development) of parents heterozygous for both markers [20]. The chi-square test was used to determine whether both alleles were transmitted together more frequently than individually. Odds ratio was used to calculate the risk of presenting the disease. Statistical significance for these tests was likewise considered for *p* < 0.05.

Gene –environment interaction analysis was carried out to determine whether environmental factors could be associated with MIH within a given polymorphism. Logistic regression analysis was used, taking into account the genotype of each polymorphism and the presence/absence of conditions described as MIH covariates in previous studies, such as premature birth, common childhood diseases, high fever, pneumonia, respiratory diseases, etc. [22,23].

## 3. Results

### 3.1. Study Sample

The final study sample consisted of 50 children with 67 parents/relatives, constituting a total of 117 cases and 100 controls.

The mean age of the kids with MIH was 9.7 years (standard deviation [SD] 2.36). Of the 50 children with MIH, 56% were boys and 44% girls. The degree of MIH was predominantly severe (58%), with moderate and mild involvement in 22% and 20% of the cases, respectively, according to the classification of Mathu-Muju (Table 1).

### 3.2. Distribution of the Allelic Frequencies

Based on the HardyWeinberg equilibrium principle, the allelic frequencies were seen to behave as expected, with a significance level of *p* < 0.05. The children with MIH presented a homozygous allele frequency of 45% (GG), a heterozygous allele frequency of 44.6% (AG), and a rare homozygous combination frequency of 10% (AA). Due to the very high sample counts, the samples had to be analyzed on two plates, generating two distribution plots. (Figure 1 and Figure 2).

### 3.3. Logistic Regression

The logistic regression analysis aimed to relate each polymorphism to the presence or absence of the factors defined as covariables or factors associated with MIH, based on previous studies. The results in this regard were inconclusive, with no evidence suggesting an alteration of the TGFBR1 gene to be related to the appearance of MIH in our study sample (*p* = 0.605; OR = 1.395 CI = 0.46–4.17).

In both the logistic regression analysis of AA versus AG + GG (*p* = 0.001) and in the regression analysis of GG versus AG + AA (*p* < 0.001), we only identified sex as a factor influencing the appearance of MIH, the latter being more frequent among boys (*p* < 0.005 in all cases referred to sex) (Table 2).

## 4. Discussion

Molar incisor hypomineralisation, according to different epidemiological studies, has a variable prevalence (depending on the studies, worldwide this varies between 2.8% to 40.2%). Very different reasons have been pointed to for this high variability: the variation in the diagnostic criteria used, the lack of calibration or standardization between reviewers, the examination conditions, and the age, social characteristics, and idiosyncrasies of the subjects included in the samples [24].

In our study we took as a reference the aforementioned oral health study carried out in Spain in 2015 [16] with a prevalence of MIH of 12% (very similar to other studies carried out in Europe, such as that of Kevredikou et al. in 2015 developed in Greece with a prevalence of 21%, or those of Jasulaytite et al. of 2007 and 2008 carried out in Lithuania and the Netherlands with a prevalence of MIH of around 14% [25,26,27]).

In all these studies the diagnostic criteria used for the categorization of MIH were those established by the European Association of Paediatric Dentistry (EAPD). Although it is true that there are other diagnostic criteria such as the enamel developmental defect (EDD) index recommended by the WHO, the vast majority of prevalence studies follow the EAPD criteria, which is why it was decided to use these diagnostic criteria in our study. It should also be taken into consideration that these EAPD criteria are found to be more prevalent than others, given that the European Academy definition also includes other diagnostic signs of MIH such as sequelae, i.e., the presence of atypical restorations or loss of molars due to MIH [2,24].

In relation to the etiology of MIH, the existing data point to a multifactorial origin underlying the clinical condition. The great clinical variability of the disorder could be related to the duration, potency, and timing of intervention of the etiological factors; hence the variability of the clinical features of MIH [28]. The most recent studies have attempted to relate MIH to genetic and epigenetic factors in genetically susceptible individuals, associated with other already investigated etiological factors.

### 4.1. Prenatal Factors

There is very little evidence that prenatal factors influence the appearance of MIH. While smoking during pregnancy and prenatal drug prescription have been shown to be associated with disorders developing during pregnancy, no such correlation has been found with respect to MIH [29]. In our study, all the considered prenatal factors showed no association with the appearance of the syndrome, though the EAPD still considers infant incubator stay and medication prescribed during pregnancy to be risk factors for MIH, with a moderate level of evidence according to the GRADE score [28,30].

Likewise, the EAPD has attributed vitamin D deficiency during pregnancy with a high level of evidence according to the GRADE, in concordance with the study published by Norrisgaard et al. These authors administered high-dose vitamin D to a group of women during pregnancy, and the results at 6 years after eruption of the first molar versus a control group not administered vitamin D clearly correlated enamel defects to a lack of vitamin D [28,31].

### 4.2. Perinatal Factors

Among the different perinatal factors evaluated, hypoxia could possibly increase the risk of MIH [30]. However, the EAPD attributes a low level of evidence to this association [28], and we likewise observed no relationship between MIH and this parameter.

Among these factors, preterm birth, low birth weight, birth complications, and caesarean section, acting alone or through a combination of them acting together, could be associated with the presence of MIH according to the systematic review conducted by Garot in 2021. This review was in fact able to demonstrate that hypoxia at birth substantially increased the likelihood of MIH [30].

In line with the above, Silva et al. reported no clear associations between perinatal factors (prematurity, low birth weight, complications at delivery, etc.) and MIH [30].

### 4.3. Postnatal Factors

The critical period for the development of MIH may possibly be found here, between the time of birth and three years of age, when amelogenesis of the permanent first molar takes place. Environmental factors such as pollution, childhood disease processes, and medications could be related to the appearance of the syndrome. However, Garot et al. recorded no etiological associations for this period [30], in coincidence with our own findings, and the EAPD attributed a low or very low level of evidence according to the GRADE score for a relationship between postnatal factors in general and the development of MIH [28].

An interesting association that is often a recurring clinical question among parents of a child affected by MIH is the use of medications in early childhood. In this regard, only antibiotics have been linked to MIH [28,30].

Other types of childhood disorders, such as certain infectious processes such as exanthematous diseases (measles, chickenpox, rubella, etc.), frequent infections in infancy such as bronchitis or otitis, or certain gastric disorders, fever, etc., or even very frequent, (depending on the environment in which the patient lives) asthmatic bronchitis, have also been linked to MIH [30].

It is known that antibiotics are prescribed to treat many of these conditions, often without absolute necessity due to overuse of these drugs. However, it is not proven whether the presence of the infectious disease rather than the use of antimicrobials is associated with MIH.

The aforementioned Garot meta-analysis of 2021 does not report any association between MIH and rubella, sinusitis, jaundice, rhinitis, malnutrition, throat infections, allergies, or diarrhea [30].

However, accepting this as a limitation of the current evidence, there seems to be a clear association between certain systemic factors and MIH in the postnatal period, given that this is when enamel maturation occurs as described in the Introduction to this article.

### 4.4. Genetic Influence

In contrast to what has been commented above, a certain body of evidence has been established relating MIH to genetic influences. In 2018, Teixeira et al. carried out a study in monozygous and dizygous twins to evaluate the possible influence of genetic factors in the origin of MIH [22]. The results showed strong concordance in the appearance of MIH in monozygous twins, indicating a clear genetic association. Epigenetic factors may possibly be implicated in the appearance of the syndrome. Although the study had methodological limitations [4,28], it opened a new path for research which we have attempted to follow in the present study.

The data of the EAPD point to a high level of evidence according to the GRADE quality-of-evidence score regarding the influence of epigenetic and genetic factors. Studies of single nucleotide polymorphisms (SNPs) in individuals with and without MIH [11,14], where SNP corresponds to a variation (polymorphism) of a single pair of bases, could explain the principles of human susceptibility to certain disease conditions. Jeremias et al. studied the genetic associations between the SNPs rs3790506 (TUFT1) and rs946252 (AMELX) and MIH, but found no correlation between these SNPs and the syndrome.

However, a later study by the same research group established a link between the SNP rs5979395 of the AMELX gene (Xq22) and MIH (odds ratio [OR] 11.7; *p* = 0.006), with 97% of the participants with MIH being found to carry the rs5979395*G allele [15].

Other authors have identified locus rs13058467, positioned near the SCUBE1 gene in chromosome 22 (*p* < 3.72 × 10^7^), as a locus possibly related to MIH [31,32]. The SCUBE1 gene plays a role in craniofacial development, and studies in animals showed it to be related to the dental papillae of the molars and incisors [33].

A genetic predisposition to MIH, together with one or more etiological factors, may play a role, since a number of authors have identified certain variants in the genes related to amelogenesis, such as ENAM, AMELX or MMP20 [11,14,31,33,34,35], or genes related to the immune response [16], in children with MIH.

The above suggests that MIH follows a multifactorial model in which genetic and/or epigenetic components play a prominent role according to recent analyses [16,36,37,38,39,40,41].

However, while the data published by Busanelli et al. in 2019 appeared to indicate a relationship between the variant of the TGFBR1 gene (related to immune response) and the appearance of MIH, we found no such association in our own study. Likewise, we observed no relationship between alterations of this gene and associated prenatal, perinatal, or postnatal parameters that could act as epigenetic factors capable of activating the mentioned gene and thus lead to the appearance of MIH.

These differences may be due to the fact that our study was population-based, in contrast to that of Busanelli et al., which was family-based. Given that family studies have greater statistical power, because they enrich the genetic effects, it is plausible that the genetic effect is more subtle, and that a much larger sample size would be needed to demonstrate a genetic association with a population-based design [42].

## 5. Conclusions

Within the limitations posed by a study of these characteristics, it can be affirmed that no relationship has been found between the TGFBR1 gene and the appearance of molar incisor hypomineralization. The main contribution of this study is that, following the line of research on genetic factors as determinants, it adds to the current research data to be taken into account in order to achieve the goal of finding its aetiology with greater certainty. More studies on this theme are needed to reach conclusions with more scientific evidence.

## Figures and Tables

**Figure 1 jpm-13-00777-f001:**
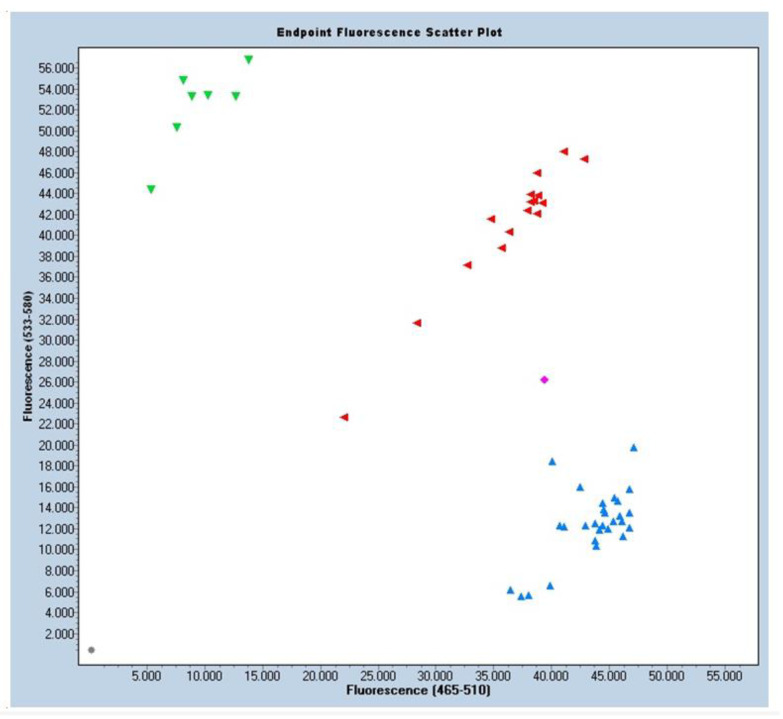
Sample distribution scatter plot. Blue fluorescence for allele G: GG. Green fluorescence for allele A: AA. Red fluorescence for heterozygosity AG. Magenta fluorescence means uncertainty, maybe because of DNA quality. Grey color is used as a negative control as it has no DNA.

**Figure 2 jpm-13-00777-f002:**
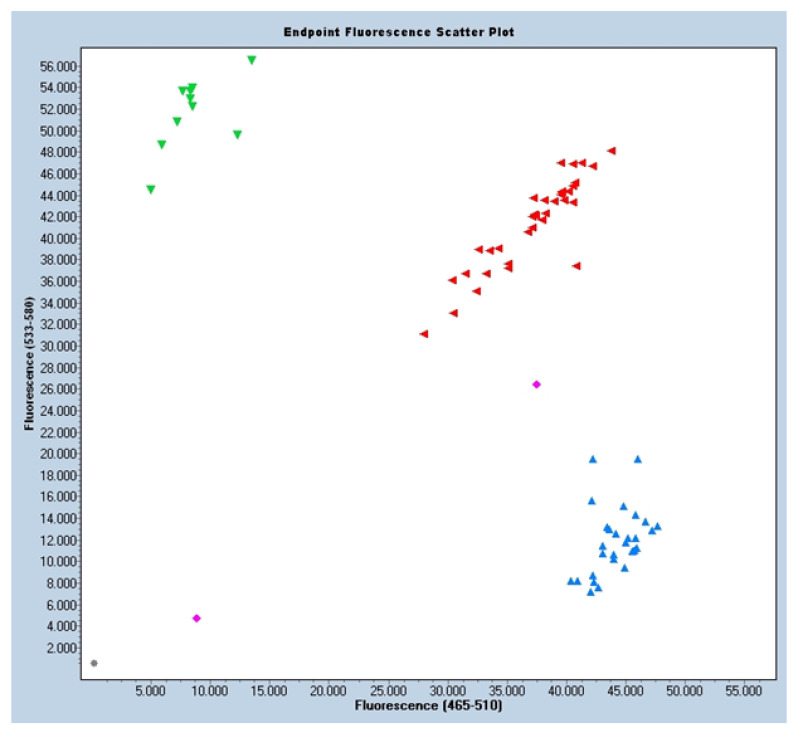
Sample distribution scatter plot. Blue fluorescence for allele G: GG. Green fluorescence for allele A: AA. Red fluorescence for heterozygosity AG. Magenta fluorescence means uncertainty, maybe because of DNA quality. Grey color is used as a negative control as it has no DNA.

**Table 1 jpm-13-00777-t001:** Sample distribution of the kids with MIH.

		Frequency	Percentage
Sex	Male	28	56%
	Female	22	44%
MIH Severity	Mild	10	20%
	Moderate	11	22%
	Severe	29	58%

**Table 2 jpm-13-00777-t002:** Logistic regression analysis applied to sex.

Logistic Regression AA vs. AG + GG Sex Value	Logistic Regression GG vs. AG + AA Sex Value
Step 0 Sig. 0.000	Step 0 Sig. 0.000
Step 1 Sig. 0.001	Step 1 Sig. 0.000
Step 2 Sig. 0.001	Step 2 Sig. 0.000
	Step 3 Sig. 0.000

## Data Availability

The data presented in this study are available on request from the corresponding author.

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
