# Peer review of "Relationship between the TGFBR1 Gene and Molar Incisor Hypomineralization"

_jpm, 2023, doi:10.3390/jpm13050777_

Round 1
Reviewer 1 Report
Please review and respond to the following comments.
Clearly, indicate the type of study design.
Remove the bullet points in Table 1 content.
What was the formula used to calculate the sample? And what values were considered to calculate the sample size? Please include this information in the corresponding materials and methods paragraph.
Analyze the data using the OR and CI. This would allow risk assessment regarding the presence or absence of the gene studied.
In the results and in the conclusion it is mentioned that there is no association between the variables studied. So the following question arises, could the presence of the gene be a protective factor and not a risk factor?
What would be the original contribution to the knowledge of this study?
Reviewer 2 Report
This study by Georgina-Perez et al., presents preliminary data on molar incisor hypomineralization. Specific comments are listed below.
- The introduction makes a great segway to draw readers' attention to this study. The aim of the study is clearly stated.
- Any result should be placed in the Result section. Remove prevalence 12.3% from line 76.
- Table 1: Since this is from another study in 2006, remove it or put it as a supplementary table. It is strange to have someone else's table as the first main table.
- Genetic analysis (lines 102-112) needs more descriptions. The current paragraph is not sufficient to explain what methods were used to present the data in the Result section. For example, Fig1 shows the range of fluorescence in both the x and y axes while the method section completely misses an explanation about it. Or reconstruct Fig1 to make a more biologically meaningful data presentation, if the fluorescence range was insignificant.
- Why number doesn't match between Table 2 and Fig1? The number of scattered triangle dots in the figure doesn't match the number in Table 2.
- Fig1 is in low resolution so it is hard to tell for sure but I see a single magenta-colored, circled dot between the red cluster and blue cluster. What is that supposed to be?
- Table 3 needs a better presentation. What are the different steps with multiple p-values? Why not show the p-value between TGFBR1 and MIH even if they are not significant? Isn't that a finding the authors aimed to find according to the Introduction section?
- The most critical pitfall of this study is the sample size. 10 or 11 in one group perhaps is not sufficient to draw a solid conclusion. This is probably the explanation for why the main finding the authors have does not match the Discussion section.
Author Response
Please, see the attachment.

Reviewer 3 Report
The paper took into account a 12.3% approach of the affected population regarding this disorder. Studies in other regions of the world have been observed in the literature. A discussion about this would be opportune.Author Response
Please, see the attachment.

Round 2
Reviewer 2 Report
With my original comment #4 regarding the genetic analysis, I do not see where and how it is improved in this version of the manuscript. Again, in the current version, the method is not fully descriptive, insufficient, and therefore not reproducible.
With my original comment #6 regarding Figure 1 image resolution, it has not been improved at all. The quality of the image is poor and not publishable. I suggest the authors submit publication-quality figures with clarity of data presentation.
With my original comment #8 regarding sample size, the authors have not addressed appropriate responses.
Round 3
Reviewer 2 Report
The manuscript has improved. I have no further comments or questions.